# Non-Motherhood between Obligation and Choice: Statistical Analysis Based on Permutation Tests of Spontaneous and Induced Abortion Rates in the Italian Context

**DOI:** 10.3390/healthcare10081514

**Published:** 2022-08-11

**Authors:** Angela Alibrandi, Lavinia Merlino, Claudio Guarneri, Ylenia Ingrasciotta, Agata Zirilli

**Affiliations:** 1Department of Economics, University of Messina, 98122 Messina, Italy; 2Papardo Hospital Company, 98158 Messina, Italy; 3Department of Biomedical, Dental and Morphological and Functional Imaging Sciences, University of Messina, 98125 Messina, Italy

**Keywords:** biostatistics, clinical epidemiology, public health, medical statistics, permutation test, standardized rates, spontaneous and induced abortion, comparison among Italian macro-areas

## Abstract

**(1) Background:** This paper aims to examine two relevant phenomena in the context of public health: spontaneous abortion (SA) and induced abortion (IA). SA is one of the most common complications of pregnancies; IA is a conscious choice that is made by the mother/couple. **(2) Methods:** Permutation tests were applied to SA and IA standardized rates detected by ISTAT (2016–2020). The NPC test, chosen for its optimal properties, was applied to compare different Italian territorial divisions (stratifying for year and age classes of women) and analyze the trend of years by stochastic ordering. **(3) Results:** Only for SA, there are significant differences among the three territorial divisions: the South records higher SA standardized rates than the North and the Center; the rates of IA are similar. Relating to distinct women age classes, the SA standardized rates do not show significant differences among the three analyzed geographical areas; different results are highlighted for IA. Stochastic ordering shows that only the IA standardized rates are characterized by a significant monotonous decreasing trend over the years. **(4) Conclusion:** The SA phenomenon has shown a decreasing trend that could be justified by the progress of science. For IA, we can certainly say that the general decrease in the phenomenon is due to the greater use of contraceptive methods that help to prevent unwanted pregnancies.

## 1. Introduction

This work analyzes abortion, both spontaneous and voluntary, under various aspects to highlight its main characteristics and outline a territorial cognitive picture in Italy over the last five years. The paper is structured as follows:

Section 1.1 presents the generalities of spontaneous abortion;

Section 1.2 illustrates the characteristics related to voluntary abortion;

Section 1.3 reviews the social and psychological aspects of abortion;

Section 1.4 contains an overview of the main scientific contributions relating to abortion (both spontaneous and voluntary);

Section 2 describes the materials and methods; in particular, the data are presented and the NPC methodology is described, with an overview of scientific articles related to application of this methodology in various fields of medical and health research;

Section 3 illustrates the results of the analyses performed by the NPC test: in particular, the first analysis was performed in order to assess the existence of significant differences between different Italian macro-areas (North, Center, and South) with reference to SA and IA using the single regions as statistical units, stratifying for year; in the second analysis, we applied the NPC test to compare the Italian macro-areas, with reference to SA and IA, stratifying for age classes (defined by ISTAT) of women in the fertility period who experience an “abortion” event; finally, in the third analysis, we assessed the changes recorded over the years, and, in particular, we verified whether there is a significant monotonous downward trend in standardized rates (referring to SA and IA), stratifying by territorial macro-areas;

Section 4 contains the discussion of the results and the working hypotheses and their interpretation, referring to previous studies;

Section 5 contains the concluding remarks, and, in addition, future developments of the research are also presented.

### 1.1. Spontaneous Abortion: Generalities

Spontaneous abortion (SA) is the most common cause of fetal death during pregnancy. According to the definition of the World Health Organization (WHO), abortion is “the expulsion or removal from the maternal organism, within 180 days of conception, of an embryo or fetus weighing 500 g or less, with a cephalous-podium length not exceeding 25 cm, and a gestational age of less than 22 weeks”.

There are several classifications relating to spontaneous abortion:-complete: when the product of conception is totally expelled;-incomplete: when the fetus is partially expelled;-internal: when the fetus is not viable and remains in the uterine cavity;-early: if it occurs before the twelfth week of gestation;-late: when it occurs between the twelfth and twentieth week of gestation;-recurrent: when two or more consecutive spontaneous abortions occur.

The etiology of spontaneous abortions is very varied. In fact, a very specific cause is often difficult to identify. The risk factors for spontaneous abortion include advanced maternal age (>35 years), chromosomal abnormalities or genetic abnormalities (abnormalities resulting from errors in both female and male gametogenesis), and immunological causes linked to alterations in the functioning of the system (this system includes various mechanisms that prevent the rejection of the product of conception). A connection has recently been highlighted between autoimmune thyroid disorders and abortion; autoimmune thyroid disease is among the most frequent causes of hypothyroidism in women of reproductive age and is often linked to both infertility and spontaneous abortion. Abortions of infectious origin have a certain relevance. This mechanism could include the formation of toxic metabolites, chronic endometrial infection, and chorioamnionitis. Viruses (herpes, cytomegalovirus, and parvovirus) seem to be the most frequently involved microorganisms, also in consideration of their potential ability to create chronic maternal infections. Systemic diseases of the mother, such as arterial hypertension, diabetes, and insulin resistance, can cause abortion. Other conditions determining abortion are developmental anomalies, or the position and formation of the uterus (the septum uterus is the most frequent malformation and is associated with the highest percentage of reproductive failures and obstetric complications). Even iatrogenic causes are of some importance, especially for the possible medico-legal implications. Prenatal diagnostic procedures (amniocentesis, carial villus sampling, and cord blood sampling) increase the risk of abortion; for amniocentesis, this risk is around 0.4%; for carial villus sampling and/or cord blood sampling, it is around 3%. The use of drugs in high doses and radiological procedures are responsible for the onset of malformations in the unborn child rather than abortion, but the intake of certain drugs, such as antiblastic ones, can certainly cause abortion. The abuse of alcohol and tobacco by women leads to abortion. In the field of occupational medicine, occupational poisoning due to lead, benzene, and mercury is a frequent cause of abortion. The abortion diagnosis is based on clinical criteria, ultrasound, and quantitative serum dosage of Beta-hCG.

### 1.2. Induced Abortion: Generalities

The voluntary termination of pregnancy, or induced or voluntary abortion (IA), consists of ending a pregnancy before the fetus can survive outside the uterine cavity. This practice is regulated differently around the world. In some countries, such as those in Latin America, it is prohibited with some exceptions if it is the only alternative to save a woman’s life. In some countries, such as those in Oceania, it is allowed but with restrictions, such as for pregnancies resulting from sexual violence or in the case of fetal malformations; in still other countries, such as most European countries, including Italy, it is allowed without particular restrictions if performed within a certain gestational age.

In Italy, voluntary termination of pregnancy is governed by Law 194 of 22 May 1978: “Rules for the social protection of maternity and on the voluntary termination of pregnancy”, confirmed by the 1981 referendum, which describes the procedure to be followed. Its primary purpose is the social protection of motherhood and the prevention of abortion. Pregnancy can be terminated for health, economic, family, or social reasons. In the first trimester of pregnancy, abortion is allowed based on a declaration by the woman who considers the pregnancy to be compromising for her physical or mental health. After the first trimester, abortion is allowed only in cases where pregnancy or childbirth involve a danger to the life of the woman, or when serious abnormalities of the fetus are found that could damage the psychophysical health of the woman.

There are two methods for carrying out voluntary termination of pregnancy: the pharmacological method and the surgical method. The surgical method, which can be performed under general or local anesthesia (in public structures of the National Health Service or private structures approved by the regions), has now been superseded by pharmacological treatment. The latter is considered the first-choice method and consists of taking drugs to terminate the pregnancy.

### 1.3. Abortion as Trauma: Psychological Consequences

Spontaneous abortion is a traumatic experience for women not only from a physical point of view but also from a psychic one [1]. The woman is upset because it is a sudden and involuntary event; this differentiates it from the voluntary interruption of pregnancy, which involves a conscious responsibility of the pregnant woman and the partner. Precisely for this reason, women who have a spontaneous abortion, despite presenting higher mental stress than women who have voluntarily interrupted their pregnancy, undergo an improvement in the initial psychological disorders more quickly. After the abortion, the woman must face a series of psychological disorders. Early pregnancy loss is known to generate symptoms of anxiety and depression in the first few weeks, which are prominent post-traumatic stress symptoms. Another symptom that prevails is a sense of guilt: the woman accuses herself of being responsible for the death of the child. Other symptoms are sadness, anger, hostility, impulsiveness, insomnia, nightmares, irritability, irrational and sudden crying, neurosis, frequent mood swings, depression, thoughts of suicide, sleep disturbances, and loss of appetite. In addition to psychological disorders, the woman could also experience sexual dysfunctions. Numerous studies underline the importance of psychological support for women and their partners who live through the experience of abortion [2,3].

The choice between an abortion and the birth of a child creates a state of emotional and cognitive disturbance in the woman. The psychological consequences of IA can occur in the short- or long-term. Initially, after performing the abortion, there is a reduction in anxiety levels since the element causing the anxiety, namely pregnancy, is removed. After the first period, many women begin to have symptoms, such as anxiety, confusion, depression, and a sense of restlessness. The symptoms that can be detected in women following the abortion are: recurrent memories of the IA experience and feelings of reliving that moment, and psychological distress due to exposure to triggers that resemble the traumatic event, such as pregnant women or newborns. Other symptoms may be: emotional disorders, communication disorders, eating disorders, thought disorders, emotional relationship disorders, sexual disorders, sleep disorders, and phobic-anxiety disorders.

### 1.4. Scientific Background

With reference to spontaneous abortion, numerous recent studies try to identify the most recurrent causes. Some studies underline the close link between the rate of spontaneous abortion and advanced maternal age [4,5,6]; others aim to identify possible causes of spontaneous abortion in the socio-economic status of women [7]; women with lower socio-economic status have a higher risk of SA, such as lower income and educational attainment, and were inversely associated with SA risk; women with agricultural and related work had a significantly greater prevalence of SA [8,9,10,11,12].

Grippo et al. 2018 aimed to investigate the association of natural abortion with environmental causes; in particular, they focused on air pollution exposure during the pregnancy period in relation to spontaneous abortion and stillbirth. A total of 43 studies were reviewed, suggesting that exposure to air pollutants (particulate matter, carbon monoxide, and cooking smoke) may cause higher risks of stillbirth and spontaneous abortion [13]; in addition, exposure to passive smoking has also been considered and identified as a possible cause of an increased risk of AS [14].

Other scientific contributions, on the other hand, focus on the risk of spontaneous abortion in professionally exposed women [15,16,17,18,19]. A study evaluating the risk of miscarriage found no differences between veterinarians exposed to inhaled anesthetics and control women [20]. However, when abortion rates were broken down into decades, the frequency observed in exposed professionals who worked in the 1970s was higher than that in the control group. In contrast, exposed professionals who worked in the 1980s had a risk of miscarriage of less than 10%, which was similar to that of the unexposed group. This result can be attributed to the types of anesthetics used over the decades and to the implantation of scavenging systems in operating theaters.

Voluntary termination of pregnancy is a major public health problem. Many scientific contributions are aimed at identifying the main causes that lead women to resort to induced abortion. In recent years, the potential effects of induced abortion on mental health have been a growing area of research [21,22]. It has been suggested that IA can have an impact on women’s relationships and can lead to a deterioration in the quality of those relationships, particularly with her partner, thus affecting the sexual sphere [23]. In many countries, the most frequent reasons that lead women to have an abortion are socio-economic concerns or the limitation of pregnancy [24].

In high-income countries, there has been a significant reduction in abortion rates in recent years; in low- and middle-income countries, on the other hand, the rates have tended to rise or remain unchanged. In Ethiopia, for example, the number of women of reproductive age seeking induced abortion is on the rise. Megersa et al., 2020 identified factors associated with induced abortion in Ethiopia [25]. The most abortive women are single, low-income women with more than two children. These findings suggest targeted government action to prevent child marriage and provide targeted sex education for adolescents.

The problem linked to complications deriving from surgical abortion procedures should not be underestimated. In Sweden, induced abortion is a very common gynecological procedure; in this background, Niinimaki et al., 2011 provide an overview of the complications related to post-abortion bacterial infections [26]. Another study found that the use of antibiotics significantly reduces the likelihood of post-abortion infections when comparing negative to positive women screened for numerous bacteria [27]. The explanation often given by women seeking an abortion is that the pregnancy was either unplanned or unwanted. However, the vast number of social, economic, and health circumstances underlying these motivations have not yet been fully examined. The reasons are often much more complex: the choice to have an abortion is usually motivated by multiple factors, some of which are attributable to the limited ability of women to control all the circumstances of their life [28].

## 2. Materials and Methods

### 2.1. NPC: Methodological Issue

The non-parametric combination (NPC) test is a recent multivariate and multistrata methodology based on permutation solution [29,30,31]. It allows to find a correct and consistent estimation of the permutation distributions, both for the partial tests and the combined tests, and to achieve effective solutions of multidimensional hypothesis testing problems in the context of non-parametric permutation inference [32]. Once a classification criterion has been established, it checks whether there are statistically significant differences between two or more groups in relation to a set of variables, measured on several statistical units [33]. Permutation tests have two main properties (assuming data exchangeability between groups):-property of similarity: whatever the distribution of the data, the probability of rejecting the null hypothesis is invariant with respect to the set of actually observed data, whatever the method of disclosure of the data;-for any α significance level, for any distribution and for all possible observed datasets, if under the alternative, the distribution dominates the null hypothesis, and then there exists an undistorted conditional test, such that the probability to reject the null hypothesis is always lower than the α significance level.

These two properties guarantee that the conditional probability of rejecting the null hypothesis (H_0_) when it is true is always equal to the significance level α, regardless of how the data are detected, and that the conditional probability of rejecting H_0_, when the alternative hypothesis (H_1_) is true, is always not less than the chosen α significance level, under the condition that data can be exchanged between groups. Thanks to these two properties, the inferences resulting in permutation tests can be extended to the entire target population, respecting the non-distortion and consistency properties [34,35].

NPC is characterized by some properties that make it preferable to other approaches of classical inference [33]:-the assumptions of normality and homoscedasticity are not required [36];-the analyzed variables can be of any nature (nominal, ordinal, and numerical);-it can be applied even when there are missing data [37];-it guarantees statistical power even in presence of low sampling size [38];-tackles problems of multivariate hypothesis testing without the need to specify the dependence structure among variables [39,40];-it offers the possibility of stratified analyses with respect to a confounding factor [41];-it allows to verify restricted alternative hypotheses (stochastic ordering) [35];-it can be applied even when the number of observed subjects is smaller than the number of variables [42].

The NPC procedure develops in two phases:-the first phase foresees the decomposition of the multivariate hypothesis system into one-dimensional sub-hypothesis, for each of which there is a partial permutation test. This allows examining the marginal contribution of each individual response variable in the comparison between the groups [43];-the second phase foresees the non-parametric combination of partial tests in a single second-order test related to the multivariate global hypothesis [44].

If, within the analysis, there is also a stratification variable, two levels of combination are foreseen, which, respectively, combine first the partial tests in combined second-order tests, each corresponding to each stratum; successively, it combines the combined tests into a single third-order combined test.

Considering the set of *p*-values for testing the respective set of partial null hypotheses, a union–intersection test considers the joint null hypothesis corresponding to a global null hypothesis that all joint null hypotheses are true; if any such partial null is rejected, the global null hypothesis is also rejected.

The NPC test uses an opportune combination function [30,31], such as Fisher, Tippett, or Liptak, when the partial tests are non-parametrically merged, through conditional Monte Carlo resampling procedure, in combined tests that are unbiased, consistent, and with asymptotic properties [45,46].

All these properties make NPC test widely flexible and broadly applicable in several fields; in particular, we cite recent applications in the context of medical research for public health protection [47,48,49,50,51,52,53,54,55]. The significance α level for all statistical analyses was 0.05 (in bold, we highlighted the significant *p*-value).

The used statistical package was NPC test, version 2.0, Statistical Software for Multivariate Nonparametric Permutation Test, Copyright 2001, Methodologica s.r.l.

### 2.2. The Data

In this paper, data from the official ISTAT source have been used; in particular, the SA and IA standardized rates were deduced for the period 2016–2020 in the Italian territory referring to regions belonging to North, Center, and South macro-areas.

SA standardized rate is “the weighted average of the specific rates for age, with weights given by the ratios between the ‘type’ population in the age groups and the total ‘type’ population, multiplied by 1000.” Similarly, IA standardized rate is “the weighted average of age-specific rates, with weights given by the ratio of the average female ‘type’ population in the age group and ‘type’ female population of childbearing age” (according to ISTAT definitions).

The rates have been standardized for each year. The absolute number of spontaneous abortions registered in Italy (source: ISTAT) is equal to 61,580 (in 2016), 55,761 (in 2017), 42,782 (in 2018), 48,932 (in 2019), and 41,493 (in 2020). The absolute number of induced abortions in Italy is 84,926 (in 2016), 80,733 (in 2017), 76,328 (in 2018), 71,642 (in 2019), and 66,413 (in 2020).

The standardized abortion rate is the most accurate indicator for a correct assessment of the SA and the appeal to the IA; it is processed with data collected, analyzed, and published by ISTAT. It is important to underline that the standardized rate is used here because it allows comparisons to be made since it takes into account the differences in the composition of the sample.

In addition, we considered the data on abortions referring to women of childbearing age (15–49 years) distinguished into following classes (according to ISTAT classification): 15–19; 20–24; 25–29; 30–34; 35–39; 40–44; 45–49 years.

Figure 1 and Figure 2 illustrate, by means of a cartogram, the SA and IA standardized rates recorded in the Italian regions in the first and last year of the examined period. These graphs make it possible to highlight the differences in these rates between the Italian regions (in the same year) and, also, to compare the two examined years for SA and IA separately.

The cartograms referring to SA highlight a greater intensity of the phenomenon in the South rather than in the North, both in the first and last year of the examined period.

Regarding IA, there is an important decrease in the maximum and minimum values from the first to the last year of the examined period, with a different distribution of regional data.

Figure 3 and Figure 4 show the boxplots of standardized rates of regional data, grouped for macro-areas (North, Center, and South) referring to SA and IA, respectively, for the period ranged between 2016 and 2020.

Observing Figure 3, we can see that SA standardized rates are higher in South throughout the examined period; North, on the other hand, is the macro-area where these rates are lower.

With an opposite trend compared to SA rates, North has the highest IA standardized rates and South the lowest (Figure 4). The phenomenon seems to have a marked decrease over the years in all territorial divisions.

## 3. Results

### 3.1. Comparison among Italian Macro-Areas, Stratifying for Year

The first analysis was performed in order to assess the existence of significant differences between different Italian macro-areas (North, Center, and South) with reference to SA and IA using the single regions as statistical units (this allows to have, within each territorial division, the replications necessary for the application of the NPC test), stratifying for year. The null hypothesis postulates the indifference between the distributions of examined variables, while the alternative one postulates the non-indifference in distributions. The hypotheses system is the following:$$H_{0i}:\;\{SA_{1i}\;\overset{d}{=}\;\ldots \;\overset{d}{=}\;SA_{3i}\}\cap \{IA_{1i}\overset{d}{=}\;\ldots \;\overset{d}{=}\;IA_{3i}\}$$
$$H_{1i}:\;\{SA_{1i}\;\overset{d}{\neq }\;\ldots \;\overset{d}{\neq }\;SA_{3i}\}\cup \{IA_{1i}\overset{d}{\neq }\;\ldots \;\overset{d}{\neq }\;IA_{3i}\}$$
where 1–3 are the three examined Italian macro-areas and i is the stratification index that is referred to years (2016–2020), so it assumes a value ranged between 1 and 5.

The hypothesis systems are expressed through the determination of partial tests (first-order) that allow to evaluate the existence of statistically significant differences between the territorial macro-areas, the combination of which allows the construction of second-order tests (combined *p*-value). In Table 1, we report mean ± standard deviations (SD) of the standardized rates of SA and IA recorded in each region belonging to the three macro-areas (North, Center, and South), stratifying by year. The last column reports the combined *p*-values obtained by using the Fisher combination function.

The results obtained from the application of the NPC test showed that, only for SA, there are significant differences among the three territorial divisions from 2017 onwards (*p* < *0.050*). On the other hand, the rates of IA are similar in the three macro-areas for each examined year.

Therefore, pairwise comparisons between macro-areas are necessary; for this analysis, the Bonferroni’s correction is applied, for which the α global error level (*0.050*) is divided by the number of pairwise comparisons (three); the adjusted significance level is *0.017.*

From the pairwise comparisons, shown in Table 2, we can see that, in all the years examined, the rates of SA in North and Center are similar; in fact, their comparison is not statistically significant. South, recording higher SA standardized rates, significantly differs from North and Center both globally (combined *p*-value < *0.017*) and for the specific years 2017 and 2019 too. In relation to the year 2020, the significance is recorded only in the North versus South comparison (*p* = *0.003*) since the Center versus South comparison is not significant after the Bonferroni’s correction.

### 3.2. Comparison among Italian Macro-Areas, Stratifying for Age Classes

In the second analysis, we applied the NPC test in order to compare the Italian macro-areas (North, Center, and South) with reference to SA and IA, stratifying for age classes of women who experience an “abortion” event; these age classes are defined by ISTAT and refer to women’s fertility period (15–49 years). In this analysis, we used the years (2016–2020) as replications according to NPC test assumptions. The hypotheses system is the following:$$H_{0i}:\;\{SA_{1i}\;\overset{d}{=}\;\ldots \;\overset{d}{=}\;SA_{3i}\}\cap \{IA_{1i}\overset{d}{=}\;\ldots \;\overset{d}{=}\;IA_{3i}\}$$
$$H_{1i}:\;\{SA_{1i}\;\overset{d}{\neq }\;\ldots \;\overset{d}{\neq }\;SA_{3i}\}\cup \{IA_{1i}\overset{d}{\neq }\;\ldots \;\overset{d}{\neq }\;IA_{3i}\}$$
where 1–3 are the three examined Italian macro-areas and i stratification index refers to age classes and assumes a value ranged between 1 and 7 (15–19; 20–24; 25–29; 30–34; 35–39; 40–44; 45–49). In Table 3, we report partial and combined *p*-values obtained by this application of the NPC test.

The results obtained from the application of the NPC test showed that the SA, referring to the distinct age classes, does not show significant differences among the three analyzed geographical areas (*p* > *0.050*). Different results are highlighted, however, in relation to IA (*p* = *0.001*), denoting significant differences among the three territorial macro-areas. We note, in fact, an evident significance referring to classes from 20 to 24 and 40 to 44 years. To understand which macro-area entails this significance, two-by-two comparisons were made, verifying restricted alternative hypotheses. Starting from the examination of the IA standardized rates shown in Figure 2 (which shows a greater use of IA in the North compared to the other two macro-areas and in the Center compared to the South), we tested by NPC test the restricted alternative hypothesis setting N > C, N > S, and C > S. Since multiple comparisons are performed, as already completed in the first analysis, it is appropriate to apply Bonferroni’s correction to check the multiplicity effect (adjusted σ = *0.017*). The results are shown in Table 4.

Observing the results obtained, we can see, for IA, a similarity condition between North and Center since all the partial and combined *p*-values are not significant (*p* > *0.017*). On the contrary, we find significance in the comparison of North versus South and Center versus South, both globally and in the single age groups; in fact, the restricted alternative hypothesis, according to which North and Center have higher IA standardized rates compared to South, is verified with reference to classes from 20 to 24 and 40 to 44 years (*p* < *0.017*).

### 3.3. Stochastic Ordering: Evaluation of a Monotone Decreasing Trend over Years, Stratifying for Italian Macro-Areas

In the third analysis, we applied the NPC test to assess the changes recorded over the years, and, in particular, we verified, by means of the multivariate and multistrata stochastic ordering procedure [35], whether there is a significant monotonous downward trend in standardized rates (referring to SA and IA), stratifying by territorial macro-areas (North, Center, and South), using regional data as a replication. We initially performed the comparisons between each year and the next (obtaining the partial *p*-values), and then the global comparison (combined *p*-value). The number of random permutations is set to 5000; the combining function for this analysis is Fisher.

Assuming that the complete dataset is divided into two pseudo-groups and that the data vectors within them are supposed to be exchangeable under the null hypothesis, the hypothesis system is:H_0*i*_: {∩_j_[∩_h_Y_h1(j)*i*_ = ^d^ Y_h2(j)*i*_]}(1)
H_1*i*_: {∪_j_[∪ _h_Y_h1(j)*i*_ ≥ ^d^ Y_h2(j)*i*_]}(2)
with *i* = 1, …, 3 (stratification index), which refers to Italian macro-areas, and j = 1, …, 5, which refers to years (2016–2020). In the hypotheses system, at least one alternative sub-hypothesis is strictly true. Table 5 shows the *p*-value obtained by the stochastic ordering procedure.

Examining the results obtained from the application of stochastic ordering (Table 5), with reference to SA standardized rates, we note that the combined *p*-value is not significant for the three Italian macro-areas; in fact, by examining the partial *p*-values, only one significance can be found for the Northern macro-area (2017 > 2018); all the remaining *p*-values do not show significance, and this indicates that the trend of SA is not characterized by a monotone ordering that decreases over the years. More interesting results, on the other hand, are highlighted in relation to the IA standardized rates, for which the stochastic ordering furnishes significant combined *p*-values for all three macro-areas; therefore, we can affirm that, throughout the Italian territory, IA is undergoing a significant decrease as it is characterized by a significant monotonous decreasing trend over the years.

## 4. Discussion

In recent years, the SA phenomenon has shown a decreasing trend that could be justified by the progress of science. Several studies have highlighted the importance of some measures that women should take to prevent the risk of abortion, such as sufficient rest, balanced diet, ban on smoking cigarettes and the use of alcoholic beverages, and limiting stress [56]. Despite this decreasing trend of spontaneous abortions, the continuing countertrend of the southern areas represents an obstacle to the definite start of the decrease in Italian statistics. The reduction in SA is certainly a symptom of the improvements in the national health system, which is added to the cases of protection of women in the working sector, with the granting of maternity leave to mothers. It is also possible to consider greater protection in daily life, guaranteed with preferential spaces and places reserved for future mothers, and, no less important, also the unspoken possibility of receiving assistance from one’s family during the nine months of gestation. Transforming these precautions into well-established social behaviors may represent the best way to ensure improvements in the gestation period. In southern society, often lacking even basic services, this task may appear difficult to carry out, but this must not be an excuse as it is necessary to build a system of social support for the future of all.

Even the psychological side of the woman affected by the early loss of the fetus should never be neglected or underestimated. All women should receive, in addition to medical assistance, psychological and relational support as well. Healthcare professionals, doctors, and midwives should combine the most typical aspects of their profession with relational aspects that are no less important [57,58]. Bereavement support should be part of clinical practice, as also highlighted by the World Health Organization.

As for IA, we can certainly say that the general decrease in the phenomenon is due to the greater use of contraceptive methods that help to prevent unwanted pregnancies. This is a consequence of the diffusion of information, on a national level, regarding responsible procreation, in which a fundamental role is played by family counseling centers. They play an important role in supporting the woman when she decides to terminate the pregnancy. The multidisciplinary skills of the professional teams, present in the counseling centers, help the woman throughout the planned pathway for IA, trying to direct her and her partner towards an informed choice. At the local level, a careful assessment must be made of numbers of counseling centers, their staff, and organization so that they can continue to play their important role. The use of the morning-after or five-day-after pill also had a significant and positive impact on the reduction in IA.

## 5. Conclusions

On the basis of the salient results obtained from the statistical analysis based on the NPC test, it is possible to state that:only for SA, there are significant differences among the three territorial divisions from 2017 onwards; on the other hand, the rates of IA are similar in the three macro-areas of Italian territory for each examined year. From the pairwise comparisons performed for only SA, in all the examined years, standardized rates of SA in North and Center are similar; South, recording higher SA standardized rates, significantly differs from North and Center both globally and for the specific years 2017 and 2019 too. In relation to the year 2020, the significance is recorded only in the North versus South comparison since the Center versus South comparison is not significant, denoting a similarity condition;the SA standardized rates, referring to the distinct age classes, do not show significant differences among the three analyzed geographical areas; different results are highlighted, however, in relation to IA, denoting significant differences among the three territorial macro-areas. We note, in fact, an evident significance referring to classes from 20 to 24 and 40 to 44 years of age. Two-by-two comparisons, which allowed to verify restricted alternative hypotheses, highlighted, for IA, a similarity condition between North and Center and a statistical significance in the comparison of North versus South and Center versus South, both globally and in the single age groups; in fact, the restricted alternative hypothesis, according to which North and Center have higher IA standardized rates compared to South, is verified with reference to classes from 20 to 24 and 40 to 44 years;the trend of SA rates is not characterized by a stochastic ordering that decreases over the years. More interesting results, on the other hand, are highlighted in relation to IA standardized rates: throughout the Italian territory, IA standardized rates are characterized by a significant monotonous decreasing trend over the years.

In the future, we aim to carry out a comparative analysis between the Italian context and other European Union countries in order to assess whether different lifestyles and different socio-economic, environmental, and religious factors can actually influence SA and IA phenomena [59]. In fact, consider Poland [60] and Malta, which are the European countries with the strictest rules on abortion. In 2020, in Poland, the Constitutional Court further limited the right to abortion, declaring “unconstitutional” the termination of pregnancy due to fetal anomalies; at the same time, an abortion is declared “legal” only when the mother’s life is in danger and in cases of pregnancies resulting from rape. In Malta, termination of pregnancy is prohibited both in the event of rape and when the mother’s life is in danger. Different situations are found in other European countries, such as Finland, Germany, and the Netherlands, which made abortion laws simpler than earlier periods.

The strength of this paper is its originality as it offers a global vision on SA and IA intensity and on their trend (over the last five years) in Italy; these two phenomena of health interest are jointly analyzed using a multivariate and multistrata statistical approach based on permutation tests. The weakness of the paper is due to the absence of statistical evaluations of psychological aspects related to SA and IA. These aspects take on great importance in the relational and social life of women who experience an abortion. For this reason, it might be desirable to deepen these aspects by means of the administration of a specific questionnaire aimed at detecting and understanding motivations, moods, and possible coping strategies implemented to deal with the stressful situation related to an abortion, whether spontaneous or induced.

## Figures and Tables

**Figure 1 healthcare-10-01514-f001:**
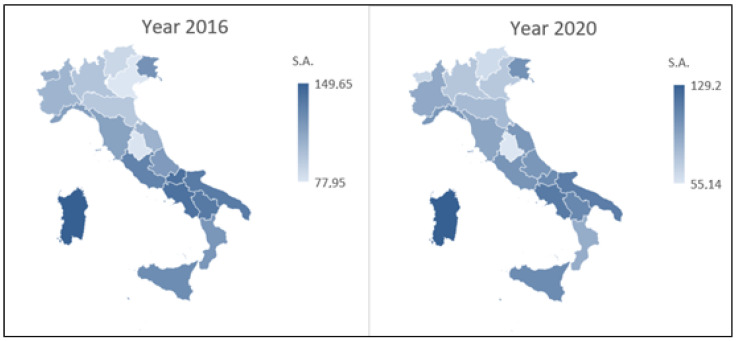
Cartogram of the Italian territory according to SA standardized rates (years 2016 and 2020).

**Figure 2 healthcare-10-01514-f002:**
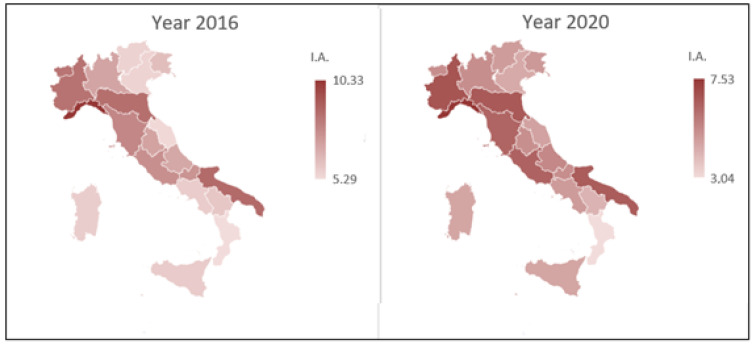
Cartogram of the Italian territory according to IA standardized rates (years 2016 and 2020).

**Figure 3 healthcare-10-01514-f003:**
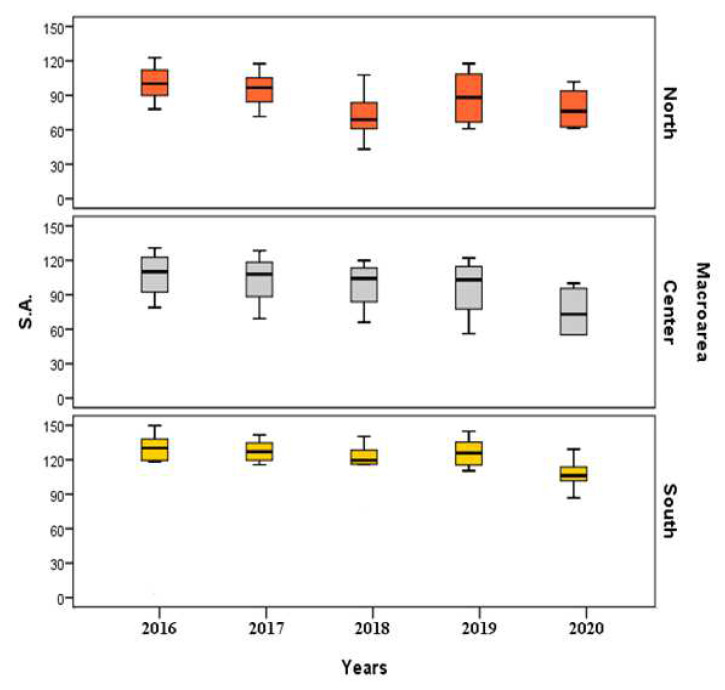
Boxplots of standardized rates of SA (2016–2020).

**Figure 4 healthcare-10-01514-f004:**
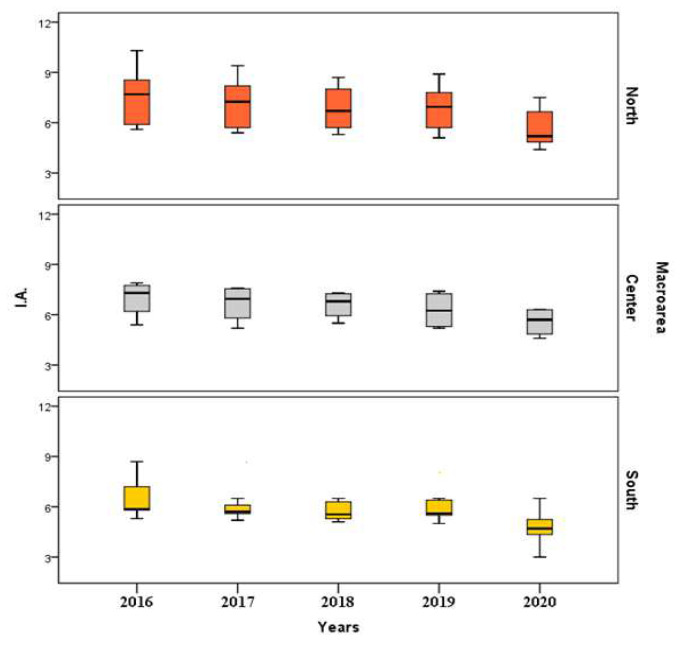
Boxplots of standardized rates of IA (2016–2020).

**Table 1 healthcare-10-01514-t001:** Mean ± SD and combined *p*-value of SA and IA standardized rates stratifying for year.

Stratum 1: Year 2016				Combined
	SA	IA		*p*-Value
North	100.6 ± 15.1	7.5 ± 1.7		
Center	107.5 ± 21.7	6.9 ± 1.1		
South	115.6 ± 47.9	6.4 ± 1.2		
	↓	↓		
*p*-value	*0.689*	*0.335*	→	*0.571*
**Stratum 2: Year 2017**				
North	95.2 ± 15.1	7.1 ± 1.5		
Center	103.4 ± 24.7	6.7 ± 1.1		
South	129.2 ± 9.0	6.2 ± 1.1		
	↓	↓		
*p*-value	** *0.001* **	*0.330*	→	** *0.003* **
**Stratum 3: Year 2018**				
North	64.6 ± 32.5	6.9 ± 1.3		
Center	98.6 ± 23.0	6.6 ± 0.8		
South	103.9 ± 45.7	5.9 ± 0.9		
	↓	↓		
*p*-value	** *0.044* **	*0.224*	→	** *0.047* **
**Stratum 4: Year 2019**				
North	92.1 ± 21.5	6.8 ± 1.0		
Center	96.1 ± 28.3	6.3 ± 1.14		
South	126.0 ± 12.2	5.9 ± 0.9		
	↓	↓		
*p*-value	** *0.005* **	*0.333*	→	** *0.011* **
**Stratum 5: Year 2020**				
North	80.9 ± 16.2	5.7 ± 1.1		
Center	82.1 ± 23.8	5.6 ± 0.9		
South	107.4 ± 12.3	4.8 ± 1.0		
	↓	↓		
*p*-value	** *0.006* **	*0.202*	→	** *0.008* **

**Table 2 healthcare-10-01514-t002:** Partial and combined *p*-values of two-by-two comparison between macro-areas, stratifying for year, for only SA.

Year	North vs. Center	North vs. South	Center vs. South
2016	*0.526*	*0.478*	*0.825*
2017	*0.485*	** *0.004* **	** *0.014* **
2018	*0.096*	*0.047*	*0.877*
2019	*0.792*	** *0.001* **	** *0.010* **
2020	*0.906*	** *0.003* **	*0.025*
	↓	↓	↓
Combined	*0.876*	** *0.001* **	** *0.011* **

**Table 3 healthcare-10-01514-t003:** Partial and combined *p*-values of comparison among macro-areas, stratifying for age classes.

Age Classes (Years)	SA	IA		
15–19	*0.057*	*0.264*		
20–24	*0.386*	** *0.006* **		
25–29	*0.598*	** *0.001* **		
30–34	*0.470*	** *0.002* **		
35–39	*0.094*	** *0.001* **		
40–44	*0.058*	** *0.007* **		
45–49	*0.065*	*0.099*		
	↓	↓		
Combined	*0.118*	** *0.001* **	→	** *0.014* **

**Table 4 healthcare-10-01514-t004:** Partial and combined *p*-values of two-by-two comparison between macro-areas, stratifying for age classes, for only IA.

Age Classes (Years)	North > Center	North > South	Center > South
15–19	*0.665*	*0.137*	*0.129*
20–24	*0.774*	** *0.016* **	** *0.002* **
25–29	*0.707*	** *0.008* **	** *0.007* **
30–34	*0.719*	** *0.006* **	** *0.006* **
35–39	*0.058*	** *0.007* **	** *0.008* **
40–44	*0.041*	** *0.016* **	** *0.009* **
45–49	*0.055*	*0.250*	*0.231*
	↓	↓	↓
Combined	*0.091*	** *0.001* **	** *0.001* **

**Table 5 healthcare-10-01514-t005:** Stochastic ordering: evaluation of a monotone decreasing trend over years.

Italian	2016 > 2017		2017 > 2018		2018 > 2019		2019 > 2020		Combined	
Macro-Areas	SA	IA	SA	IA	SA	IA	SA	IA	SA	IA
North	*0.237*	** *0.031* **	** *0.038* **	** *0.034* **	*0.267*	*0.985*	*0.294*	** *0.036* **	*0.334*	** *0.029* **
Center	*0.381*	** *0.048* **	*0.346*	*0.471*	*0.852*	** *0.047* **	*0.285*	** *0.033* **	*0.140*	** *0.014* **
South	*0.289*	** *0.042* **	*0.059*	** *0.029* **	*0.217*	*0.847*	*0.054*	** *0.010* **	*0.449*	** *0.023* **

## Data Availability

The data used for this work are available on the ISTAT website under the heading *Italian Statistical Yearbook*, “Health” section.

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
