# Peer review of "Non-Motherhood between Obligation and Choice: Statistical Analysis Based on Permutation Tests of Spontaneous and Induced Abortion Rates in the Italian Context"

_healthcare, 2022, doi:10.3390/healthcare10081514_

Round 1

Reviewer 1 Report

I read with great interest the Manuscript titled “Non-motherhood between obligation and choice. Statistical analysis based on permutation tests of spontaneous and induced abortion rates in the Italian context” (healthcare-1840881), which falls within the aims of Healthcare.    

In my honest opinion, the topic is interesting enough to attract the readers’ attention. The methodology is accurate and conclusions are supported by the data analysis. Nevertheless, the authors should clarify some points and improve the discussion by citing relevant and novel key articles on the topic.

Authors should consider the following recommendations:

-       Manuscript should be further revised by a native English speaker.

-       The authors have not adequately highlighted the strengths and limitations of their study. I suggest better specifying these points.

-        In the conclusions of their study, the authors write that it would be interesting to conduct a comparative analysis between the Italian context and other European Union Countries. In this regard, it would be interesting to report already in this paper some differences between Italy and European countries with stricter abortion legislation, such as Poland, especially regarding the influence of socio-cultural factors on the choice to have an abortion. Some interesting studies on the topic are the following: DOI: 10.1080/13625187.2020.1783652; DOI: 10.3390/ijerph16183413.

Author Response

We thank you for the valuable suggestions that have allowed us to improve the quality of our article.

  • Our manuscript has been revised by an English expert.
  • In the last section of our paper we added the strengths and the limitations of our paper
  • In the conclusions of our study, in according to your suggestion, we focused our attention on other Europeans Countries, such as Poland, Germany, Netherland etc… regarding the influence of socio-cultural and religious factors on the choice to have an abortion. These considerations represent for us a stimulating point of interest for future developments. In addition we cited the references that you have suggested (DOI: 10.1080/13625187.2020.1783652; DOI: 10.3390/ijerph16183413).

Reviewer 2 Report

This attudy mainly tries to have a cmparison of tw trends. Therefore it could be have designed a t test in advance to the method that you mainly have used in this study. 

Author Response

The statistical method we used in this study represents the methodologically most suitable solution to provide the answers to the questions we have initially posed. In particular, we needed a multivariate and multistratum methodology, free from parametric assumptions, since we analyze standardized rates (referred to S.A. and I.A.).

Furthermore, the use of the NPC test is widely spread and tested since 2001 in various research fields, precisely because of its flexibility and its numerous advantages (it allows to verify restricted alternative hypotheses, it is powerful even in cases of low sample size, etc.)